# Risk Factors of the First-Time Stroke in the Southwest of Saudi Arabia: A Case-Control Study

**DOI:** 10.3390/brainsci11020222

**Published:** 2021-02-11

**Authors:** Adel A. Alhazzani, Ahmed A. Mahfouz, Ahmed Y. Abolyazid, Nabil J. Awadalla

**Affiliations:** 1Neurology Division, Department of Internal Medicine, College of Medicine, King Saud University, Riyadh 11451, Saudi Arabia; aalhazzani2@ksu.edu.sa; 2Departments of Family and Community Medicine, College of Medicine, King Khalid University, Abha 61421, Saudi Arabia; drzizous2000@yahoo.com (A.Y.A.); njgirgis@kku.edu.sa (N.J.A.); 3Department of Epidemiology, High Institute of Public Health, Alexandria University, Alexandria 21511, Egypt; 4Department of Community Medicine, College of Medicine Mansoura University, Mansoura 35516, Egypt

**Keywords:** first-time stroke, risk factors, matched multicenter case-control, southwest Saudi Arabia

## Abstract

Worldwide, stroke is the second leading cause of death and a frequent cause of permanent disability. The objective was to identify the first-time stroke modifiable risk factors in the Aseer region, southwest Saudi Arabia. In a multicenter hospital-based case-control study design, all first stroke patients admitted to hospitals in the Aseer region were included consecutively over one year. Age, sex, and geographical residence-matched controls were selected and included in a ratio of 1:1. Data collected included altitude (low or high), nationality, marital status, educational level, family history of stroke, history of diabetes mellitus, history of systemic hypertension, high cholesterol level, current smoking, obesity, and regular exercise practice. The study included 1249 first-time stroke patients and 1249 age, sex, and residence-matched controls. Hypertension, diabetes mellitus, obesity, and high cholesterol were significantly highly prevalent among cases (57.7%, 49.4%, 42.0%, 29.4%, respectively) compared to controls (31.8%, 25.9%, 30.8%, 12.1%, respectively). Practicing regular exercise was significantly highly prevalent among controls (29.9%) compared to cases (13.1%). Multivariable logistic regression analysis revealed that systemic hypertension (adjusted odds ratio (aOR) = 2.12, 95%CI: 1.74–2.57), diabetes mellitus (aOR = 1.73, 95%CI: 1.41–2.21), obesity (aOR = 1.95, 95%CI: 1.61–2.28) and high cholesterol (aOR = 1.64, 95%CI: 1.28–2.10) were significant risk factors, while regular exercise practice was a significant protective factor (aOR = 0.12, 95%CI: 0.05–0.26) for stroke. Hypertension, diabetes mellitus, obesity, and high cholesterol are major risk factors for stroke in the Aseer region of southwest Saudi Arabia. The protective role of regular physical activity in reducing the risk of stroke is evident. The observed higher prevalence of potentially modifiable risk factors among stroke cases encourages an urgent need to develop and implement a national program to control these factors.

## 1. Introduction

Worldwide, stroke is a significant public health burden. Stroke remains the second leading cause of death worldwide, with 5.5 million (95% confidence interval of 5.3–5.7) deaths attributed to this cause in 2016. Because of the growing size and aging of the world’s population, the global stroke burden is increasing dramatically [1]. A study showed that stroke is no longer a disease of the rich and developed world. It showed that low and middle-income countries accounted for 86 percent of global stroke deaths [2].

Research on the main risk factors for stroke in developed countries is well established, and there is growing evidence that these factors are similar in low and middle-income settings. The growing incidence of stroke in developing countries, including Saudi Arabia, is significantly related to certain modifiable risk factors including: sociodemographic, lifestyle, and behavioral factors that influence health conditions such as hypertension, diabetes mellitus, hyperlipidemia, and cardiac disease [3].

A wide range of lifestyle and behavioral factors, including a sedentary lifestyle, obesity, and stress, influence stroke incidence, either directly or by affecting other risk conditions, such as blood pressure or diabetes. A study documented that the relative risk of cerebral infarct associated with cigarette smoking was 1.9; the increased risk was higher when comparing current smokers vs. never and past smokers [4]. Obesity has a significant effect both directly and through predisposing conditions such as hypertension and diabetes [5].

A meta-analysis of observational data for physical activity and stroke reveals that lack of physical activity is a modifiable risk factor for both total stroke and stroke subtypes [6].

A range of different health conditions significantly increases the risk of stroke. A recent review article has validated that high blood pressure is the most important modifiable risk factor of stroke, increasing stroke risk four to six times [7]. Other chronic conditions, such as diabetes, heart conditions, and high cholesterol, also increase stroke risk. Studies from developing countries show that diabetes increases the risk twofold, while arterial fibrillation increases the risk six times [8].

Many studies across Saudi Arabia reported similar predisposing health conditions associated with an increased risk of stroke. Studies across the kingdom found hypertension and diabetes to be the major predisposing health conditions of stroke [9]. In Hofuf, 40.4% of stroke patients had hypertension with diabetes, while 24.9% had hypertension alone, and 11.6% had diabetes mellitus alone [10]. A similar trend was observed in Gizan. Major risk factors included hypertension (45.6%), heart diseases with or without atrial fibrillation (1.1%), and diabetes mellitus (22.8%) [11].

The Aseer region is located in the southwest of Saudi Arabia, covering more than 80,000 km^2^. The region extends from the high mountains of Sarawat (with an altitude of 3200 m above sea level) to the Red Sea and lies a few kilometers from the northern border of neighboring Yemen. The region borders Jizan and is located to its northeast. Recent data regarding the first-stroke risk factors in the Aseer region, in particular, are scarce and even lacking. Additionally, most of the previous studies in Saudi Arabia were from a single hospital. The present work aimed to study the first-time stroke risk factors in the Aseer region and review relevant literature published in other regions of Saudi Arabia.

## 2. Materials and Methods

### 2.1. Study Setting

The last reported Aseer region population was 2,166,983 (Demographic survey 2016, The Saudi General Authority for Statistics) [12]. Health services delivery in the region is provided by 242 primary health care centers, 12 secondary care hospitals, and one tertiary hospital (Aseer central hospital).

### 2.2. Study Design

A multicenter, matched, and hospital-based case-control study design.

### 2.3. Study Population

All first stroke patients admitted to all hospitals in the Aseer region over one year (January through December 2016) were included consecutively. Stroke patients outside the Aseer region were excluded from the study. Cases were diagnosed and confirmed by neurologists based on the Saudi Ministry of Health criteria (MOH Pocket Manual in Critical care and ICD-10). The pathological subtype of stroke was ischemic stroke in 1117 subjects (89.4%) and intracerebral hemorrhage in 132 subjects (10.6%).

Age and sex-matched controls were selected from patients admitted to the same hospitals during the same period. The case-control ratio was 1:1.

### 2.4. Data Collection

Data were collected by a standard questionnaire, which was developed by using the WHO STEPwise approach to chronic disease risk factors surveillance [13]. Data collected included altitude (low or high), nationality (Saudi or Non-Saudi), marital status, educational level, family history of stroke, history of diabetes mellitus, history of systemic hypertension, high cholesterol level, current smoking, obesity, and regular practice of exercise.

Hypertension was considered by either history of hypertension or the composite of self-reported hypertension, or a blood pressure measurement of 140/90 mm Hg or higher. Diabetes mellitus was defined by either a self-reported history of diabetes mellitus or HbA1c of 6.5% or higher. Regular physical exercise was defined as any regular engagement in physical activities such as running, cycling, swimming, brisk walking, playing football, etc. Smoking status was defined as currently being a smoker or currently being a non-smoker. Body mass index (BMI) was estimated based on measured weight and height. Obesity was considered when BMI was 30 kg/m^2^ or higher. Hypercholesterolemia was considered when a patient had a diagnosis of it and/or was on lipid-lowering agents or had total fasting blood cholesterol of >200 mg/dL (5.3 mmol/L) in the hospital stay.

### 2.5. Data Analysis

Data were analyzed using SPSS version 22. Frequencies and proportions were calculated. Crude odds ratio (cOR) and concomitant 95% confidence intervals (95%CIs) were computed to compare cases and controls. Multivariable logistic regression analysis was used. All significant variables in the univariate analysis were entered into the model. Adjusted odds ratio (aOR) and concomitant 95% CIs were calculated. The model used was Forward Walid [14].

## 3. Results

The present study included 1249 first-time stroke patients and 1249 age and sex-matched controls admitted to the study hospitals during the study period. Stroke patients were 776 males and 473 females. Regarding their age, 432 patients were less than 60 years of age, 496 were aged between 60 and 79, and 321 were 80 years old or above.

Table 1 shows the distribution of cases and controls by sociodemographic, family, and medical conditions. The study showed that 71.4% of cases were married, compared to 72.7% among controls. The difference was not statistically significant (cOR = 1.07, 95%CI: 0.89–1.13). Other sociodemographic factors (nationality and level of education) were not significantly different among cases and controls. Similarly, family history of stroke and consanguinity were not significantly different among cases and controls.

Regarding medical conditions, the study revealed that people with diabetes have significantly more than two times (cOR = 2.80, 95%CI: 2.36–3.31) the chance to have a stroke compared to non-diabetics. The following significant risk factors were also identified: having systemic hypertension (cOR = 2.93, 95%CI: 2.48–3.45), high cholesterol level (cOR = 3.30, 95%CI: 2.45–3.73), and obesity (cOR = 1.63, 95%CI: 1.38–1.92). Hypertension, diabetes mellitus, obesity, and high cholesterol were significantly highly prevalent among cases (57.7%, 49.4%, 42.0%, 29.4%, respectively) compared to controls (31.8%, 25.9%, 30.8%, 12.1%, respectively).

Current smoking was not significantly different among cases and controls. On the other hand, practicing regular exercise was a significant protective condition among cases and controls (cOR = 0.35, 95%CI: 0.29–0.43). Practicing regular exercise was significantly highly prevalent among controls (29.9%) compared to cases (13.1%).

All significant variables were included in a multivariable logistic regression analysis (Figure 1). After adjusting the variables for each other, the study showed that systemic hypertension, diabetes mellitus, high cholesterol, and obesity were significant risk factors. At the same time, regular exercise was a protective factor for stroke. 

## 4. Discussion

The present study in the Aseer region, Saudi Arabia, identified systemic hypertension, diabetes mellitus, high cholesterol, and obesity as significant risk factors. At the same time, regular exercise was a significant protective factor for the first stroke.

Hypertension among first stroke patients in Aseer was a significant independent predictor. It was the highest observed risk factor, and it amounted to 57.7%. Studies in different regions in Saudi Arabia reported similar figures. The figure ranged from 29.6% in Taif [15], 43.2% in Hail [16], and 73.9% in Qassim [17]. These studies found that hypertension is the most important risk factor for stroke among the Saudi population. A review article also reported systemic hypertension as the most predominant risk factor among stroke patients in the Middle East, observed in 24.9–80% of stroke patients [18]. Similarly, worldwide, a strong correlation between stroke and systemic hypertension is found by several studies. Moreover, the evident interaction between hypertension and other risk factors, including dyslipidemia, diabetes, obesity, and smoking, strongly increases the cumulative risk for stroke in patients with hypertension [19].

Saudi Arabia suffers from a high prevalence rate of hypertension among the adult population, reaching 26.1% [20]. Additionally, the adult population showed a low awareness of hypertension as a risk factor for stroke [21] Our present study showed that one-third of the control group were hypertensive. Surprisingly, a study reported that only 23% of hypertensive patients in Saudi Arabia are aware of their blood pressure elevation [9]. Thus, strict primary and secondary preventive measures of hypertension are urgently recommended to minimize the stroke burden [22].

In the present study, diabetes mellitus was the second reported risk factor for stroke, present in 49.4% of the patients. Additionally, about one-fourth of the control group were diabetic, which is an alarming figure. Studies in different regions in Saudi Arabia reported parallel figures. The figure ranged from 34.6% in Taif [15], 35.0% in Dammam [23], 42.7% in Hail [16], 57.0% in Qassim [17], and 59.0% in Madinah [24]. A review article in the Middle East also reported diabetes mellitus as the second most prevalent risk factor among stroke patients in the Middle East countries, observed in 5.1–69.4% of stroke patients [18]. Diabetic patients usually have vascular complications and additional hazards such as hypertension, dyslipidemia, and obesity that exaggerated the risk of stroke [25]. Globally, more than 415 million cases of diabetes mellitus pose a high risk for cardiovascular diseases, including stroke [26].

The WHO reported that Saudi Arabia is the second-highest country in the Middle East, and seventh globally, for diabetes. Around 7 million of the population are diabetic, and almost 3 million are prediabetic. The most worrying is that Saudi Arabia registered a ten-fold increase in diabetes in the past thirty years. Therefore, the health burden due to diabetes, including stroke, in Saudi Arabia is expected to rise to a serious level unless an effective control program is implemented [27]. Moreover, diabetes is undiagnosed in around 28% of cases, of which around half already have vascular complications [28]. Moreover, the adult population in the Aseer region showed a low awareness (34.9%) of diabetes as a risk factor for stroke [21]. Therefore, a comprehensive diabetes control program, including primary and secondary preventive measures, will minimize the personal and health impacts of diabetes [29].

The third prevalent independent risk factor of stroke revealed in the present study was obesity, observed in 42.0% of cases. Studies in other regions in Saudi Arabia reported similar figures. A review of prevalence of obesity in Saudi Arabia, the figure fluctuated from 16.7% in Taif [15], 38.1% in Riyadh [30], 42.5% in Qassim [17], to 75.0% in Madinah [24]. A systematic review study also reported obesity as a prevalent risk factor of stroke in the Middle East countries, observed in 5.3–66.0% of stroke patients [18]. In Saudi Arabia, studies indicate a rising trend in the prevalence of obesity and overweight. This increase is associated with an increased risk of many other diseases, including hypertension, diabetes, sleep problems, hypercholesterolemia, and vascular diseases [31]. One out of each three women and one out of each four men are obese. A sedentary lifestyle and high energy diet increase the risk [32]. Unfortunately, most of the adult population (66.0%) in the Aseer region are unaware of the risk of obesity for stroke [21].

In the current study, high blood cholesterol was the fourth predominant independent risk factor for stroke, reported in 29.4% of the cases. Studies in different regions in Saudi Arabia reported equivalent figures. The figure ranged from 14.6% in Hail [16] 15.3% in Qassim [17], 43.5% in Dammam [23], 49.1% in Riyadh [30], 63.0% in Taif [15], and 70.0% in Madinah [24]. A systematic review article also reported dyslipidemia as a predominant risk factor of stroke in the Middle East, observed in 5.4–65.8% of stroke patients [18]. Unfortunately, high cholesterol is a prevalent problem among the Saudi population; a population-based study revealed that 54.9% and 53.2% of adults have hypercholesterolemia [33].

A promising result in the present study is the observed protective effect of regular physical exercise against stroke. A study in Madinah, Saudi Arabia, observed a similar result [24]. Being physically active increases vascular integrity by different methods producing positive effects on risk factors for stroke. For example, physical activity has a beneficial effect on reducing blood pressure and body weight, and controlling blood lipids and diabetes [34]. In a meta-analysis study, high physical activity reduced 25% of the stroke risk than low activity in cohort studies and 64% in case-control studies [35]. The WHO reported that 58.5% of the adult Saudi population are physically inactive (52.1% of men and 67.7% of women) [36]. Moreover, awareness about the stroke risk of physical inactivity is too low (27.0%) among the adult Saudi population, especially in the southwest region [21].

The present results were in line with the Global Burden of Disease Study 2013 [37] and the global and regional case-control study (INTERSTROKE 2007–2015) [38] which concluded that the global stroke burden is mostly attributable to potentially modifiable risk factors. Out of the ten modifiable risk factors of stroke identified in the INTERSTROKE study, our study confirmed the significant contribution of five modifiable risk factors: hypertension, diabetes mellitus, physical inactivity, obesity, and dyslipidemia. The odds ratio of hypertension in the present study was equal to 2.12, while the INTERSTROKE study was 2.56; similarly, for Diabetes Mellitus (OR = 1.73 in our study vs. 1.16 in INTERSTROKE), and regular physical exercise (OR = 0.12 vs. 0.60 in INTERSTROKE). Regarding obesity, we used BMI ≥ 30.0 rather than the waist–hip ratio in INTERSTROKE, and OR was 1.95 vs. 1.44 in INTERSTROKE. For dyslipidemia, we used high cholesterol as an indicator of dyslipidemia rather than apolipoproteins B/A ratio in the INTERSTROKE study, and OR was 1.64 in our study vs. 1.84 in INTERSTROKE study.

Although the prevalence of current smoking was higher among cases (10.3%) than controls (9.0%), it was not significantly associated with stroke. Further analysis of the smoking status by duration and number of cigarettes per day may be necessary to properly examine this association. The variation between the results of the present study and the INTERSTROKE study could be related to variation in the selection of control subjects (hospital based or community based), method of assessment of the risk factor, and differences in the prevalence of risk factors among populations of different regions [38].

The present study’s strengths are derived from the large number of first stroke cases recruited prospectively during the study period. Additionally, the design being a matched case-control enhanced the validity of the results. Additionally, the study covered all hospitals in a region of Saudi Arabia. This representative wide sample allowed the generalization of the study results. On the other hand, limitations include being a study in one region of Saudi Arabia, and only the patients admitted to the region’s hospitals were included. It was challenging to study alcohol abuse effect in the present study due to cultural and religious reasons. Moreover, the association of cardiac causes, including atrial fibrillation, with stroke was difficult to examine due to the difficulty in their assessment among controls. Further national based studies are needed.

## 5. Conclusions

The present study has clearly shown that hypertension, diabetes mellitus, obesity, and hypercholesterolemia are major risk factors for stroke in the Aseer region of southwest Saudi Arabia. The current study highlighted the protective role of regular physical activity in reducing the risk of stroke. The observed higher prevalence of potentially modifiable risk factors, including hypertension, diabetes mellitus, obesity, high cholesterol, and physical inactivity among stroke cases, encourages an urgent need to develop and implement a national program to control these factors. This program should emphasize improving public awareness and applying interventions to minimize the emergence of these factors and to improve control of them.

## Figures and Tables

**Figure 1 brainsci-11-00222-f001:**
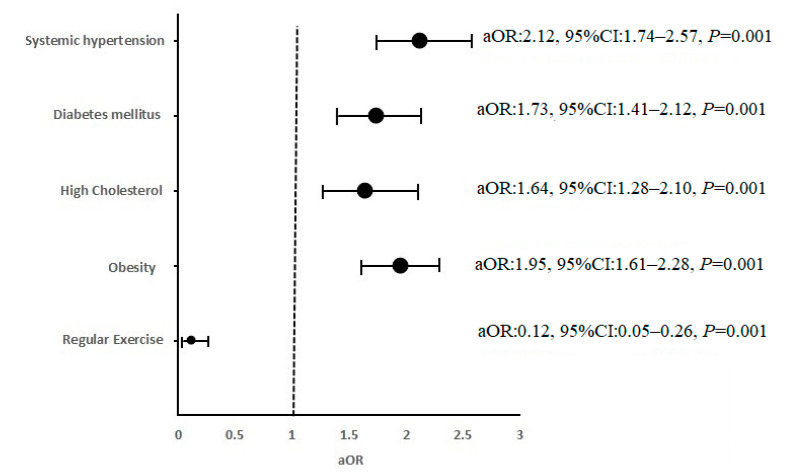
Forest plot showing the independent predictors for first-time stroke in the Aseer region.

**Table 1 brainsci-11-00222-t001:** Univariate analysis of factors associated with first-time stroke in the Aseer region.

Factors		Cases *N* (%)	Controls *N* (%)	cOR (95%CI)
Nationality	Saudi	1157 (92.6)	1170 (93.7)	Ref.
	Non-Saudi	92 (7.4)	79 (6.3)	0.85 (0.62–1.15)
Marital Status	Married	892 (71.4)	908 (72.7)	Ref.
	Single/widowed	357 (28.6)	341 (27.3)	1.07 (0.89–1.13)
Education	Illiterate	576 (46.1)	524 (42)	Ref.
	Primary	376 (30.1)	367 (29.4)	0.93 (0.77–1.12)
	Intermediate	114 (9.1)	145 (11.6)	0.72 (0.54–0.98)
	Secondary	106 (8.5)	115 (9.2)	0.84 (0.63–1.12)
	University	77 (6.2)	98 (7.8)	0.71 (0.52–0.99)
Consanguinity	No	706 (56.5)	707 (56.6)	Ref
	Yes	543 (43.5)	542 (43.4)	1.00 (0.86–1.18)
Family History	No	1107 (88.6)	1133 (90.7)	Ref.
	Yes	142 (11.4)	116 (9.3)	1.25 (0.97–1.62)
Diabetes mellitus	No	632 (50.6)	926 (74.1)	Ref.
	Yes	617 (49.4)	323 (25.9)	2.80 (2.36–3.31)
Systemic hypertension	No	528 (42.3)	852 (68.2)	Ref.
	Yes	721 (57.7)	397 (31.8)	2.93 (2.48–3.45)
High cholesterol	No	882 (70.6)	1098 (87.9)	Ref.
	Yes	367 (29.4)	151 (12.1)	3.30 (2.45–3.73)
Current Smoking	No	1121 (89.7)	1136 (91.0)	Ref.
	Yes	128 (10.3)	113 (9.0)	1.15 (0.88–1.50)
Regular Exercise	No	1085 (86.9)	875 (70.1)	Ref.
	Yes	164 (13.1)	374 (29.9)	0.35 (0.29–0.43)
Obesity	No	724 (58.2)	864 (69.2)	Ref.
	Yes	525 (42.0)	385 (30.8)	1.63 (1.38–1.92)

cOR—crude odds ratio, 95%CI—95% confidence interval, Ref.—reference group.

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
