# Peer review of "Risk Factors of the First-Time Stroke in the Southwest of Saudi Arabia: A Case-Control Study"

_brainsci, 2021, doi:10.3390/brainsci11020222_

Round 1
Reviewer 1 Report
The paper can add some new information about risk factors in different regions. Author have said that it is a first such the study in Aseer region of southwest Saudi Arabia.
Of course, such information is important for the region.
Authors have included to the study only patients admitted to hospital. Counting incidence rate, it looks rather low. Authors do not mention which approximately percent of stroke patients in this region are admitted to the hospital during one year. It is important, maybe it is only selected group of strokes in the region. It should be in study limitation.
There is no information about stroke type, at least ischemic or hemorrhagic.
There are variability between countries and regions in distribution of stroke risk factors. In the present work authors do not mention important papers published in last decade by O’Donnel group about methodology and results of Interstroke study (Neuroepidemiology 2009, Lancet 2010 and 2017). Saudi Arabia participated in this study. Differences between regions in stroke epidemiology were also discussed in many papers by Feigin. Own results should be compared to Interstroke study, in which Aseer region probably did not participate.
Author Response
The authors would like to thank the reviewer for the comments. Changes are made in red color in the attached file.

Reviewer 2 Report
In this original article entitle “Risk factors of the first-time stroke in the southwest of Saudi Arabia: A case-control study”, Alhazzani et al. presented the first multicenter hospital-based case-control study performed in the southwest of Saudi Arabia population.
In this study authors found hypertension, diabetes mellitus, obesity and high cholesterol levels to be independent risk factors in the Saudi Arabia. The presented risk factors are well stablished stroke risk factors in other populations. However, its novelty lacks in the validation of those risk factors in the Saudi Arabia population, which it is not been included in other populations studies.
Furthermore, authors found that regular physical activity is a protective factor against stroke risk. This protective effect of regular exercise has been previously reported in other studies using smaller sample sizes, which have made this association questionable. Thus, this study validates the protective effect of regular physical activity against stroke risk in a paired case-control study, making the results reliable. This finding represents a new easy way to reduce the risk of stroke.
Finally, this article provides a summary of the main research conducted on the risk of first stroke in the Saudi Arabia population, providing a good reference for readers and future studies.
Despite the remarkable findings of this article, there are some minor issues that I would like to report:
- One additional well stablished stroke risk factor in other populations is Atrial fibrillation. Despite the authors comment in the introduction that atrial fibrillation has been identified in other studies, they do not include this variable in the data collected. I consider that it could be interesting to include it in order to further validate the independent association of the other variables found; focus on the protective factor ‘regular physical activity’.
- It is not specified in the manuscript the cut-off values considered for any medical or sociodemographic values. Authors should include the criteria used as it can vary from other previous studies and it could be useful for future studies.
- Even the cOR are include for each variable, I would recommend to include the p-values obtained in the multivariable regression analysis in order to show clearly its independent association.
- The proportion of the main types of stroke (ischemic and hemorrhagic) is not specified in the study. As these strokes can have different risk factors, I considered that it could be interesting to verify if the protective effect of regular exercise is maintained when the sample is divided into these subtypes.
Taking all into account, I considered that this manuscript could be a good contribution for the Brain Sciences journal.
Author Response

(The authors gave the same response as above.)

Round 2
Reviewer 1 Report
Propose to delate new sentence line 155-159, as analyzing other risk factors authors are not performing it according to stroke subtype.